# Dietary Energy Levels Affect Rumen Bacterial Populations that Influence the Intramuscular Fat Fatty Acids of Fattening Yaks (*Bos grunniens*)

**DOI:** 10.3390/ani10091474

**Published:** 2020-08-22

**Authors:** Rui Hu, Huawei Zou, Hongze Wang, Zhisheng Wang, Xueying Wang, Jian Ma, Ali Mujtaba Shah, Quanhui Peng, Bai Xue, Lizhi Wang, Suonan Zhao, Xiangying Kong

**Affiliations:** 1Low Carbon Breeding Cattle and Safety Production—University Key Laboratory of Sichuan Province, Animal Nutrition Institute, Sichuan Agricultural University, Chengdu 611130, China; 14648@sicau.edu.cn (R.H.); zhwbabarla@126.com (H.Z.); mujtaba43@gmail.com (H.W.); wangxuey_91@163.com (X.W.); Crazyma0411@163.com (J.M.); alimujtabashah@sbbuvas.edu.pk (A.M.S.); pengquanhui@126.com (Q.P.); xuebai68@163.com (B.X.); wanglizhi08@aliyun.com (L.W.); 2Animal Husbandry and Veterinary Institute, Haibei 812200, China; fengpli@126.com (S.Z.); ruitianhu@yeah.net (X.K.)

**Keywords:** yak, dietary energy, rumen bacteria, intramuscular fatty acids, 16S rDNA sequencing, association analysis

## Abstract

**Simple Summary:**

Yak, a bovid animal, is the predominant livestock on the Qinghai–Tibet Plateau. Rumen is an important digestive organ for ruminants, such as cattle, yak, and sheep. Rumen bacteria play a crucial role in dietary energy digestion of yaks and in their adaptation to the plateau environment. Dietary energy levels affect rumen bacterial populations and lipid deposition in the meat of ruminants. The intramuscular fat fatty acid profile is important for meat quality and human health. This study was conducted to determine the rumen bacterial populations affected by dietary energy levels and understand their relationship with intramuscular fat fatty acids. The results found that increasing dietary energy significantly increased ruminal propionate concentration and reduced the ammonia concentration. High dietary energy increased the ratio of *Firmicutes* to *Bacteroidetes* and mainly increased ruminal amylolytic and propionate-producing bacteria populations. Ruminal acetate and propionate were positively related to intramuscular saturated fatty acid content, and *Prevotella* was positively related to intramuscular polyunsaturated fatty acid content and negatively related to intramuscular saturated fatty acid content. This study gives insights into how the effects of dietary energy on rumen bacterial populations are related to intramuscular fat fatty acids of yaks.

**Abstract:**

The yak rumen microflora has more efficient fiber-degrading and energy-harvesting abilities than that of low-altitude cattle; however, few studies have investigated the effects of dietary energy levels on the rumen bacterial populations and the relationship between rumen bacteria and the intramuscular fatty acid profile of fattening yaks. In this study, thirty yaks were randomly assigned to three groups. Each group received one of the three isonitrogenous diets with low (3.72 MJ/kg), medium (4.52 MJ/kg), and high (5.32 MJ/kg) levels of net energy for maintenance and fattening. After 120 days of feeding, results showed that increasing dietary energy significantly increased ruminal propionate fermentation and reduced ammonia concentration. The 16S rDNA sequencing results showed that increasing dietary energy significantly increased the ratio of *Firmicutes* to *Bacteroidetes* and stimulated the relative abundance of *Succiniclasticum*, *Saccharofermentans*, *Ruminococcus*, and *Blautia* populations. The quantitative real-time PCR analysis showed that high dietary energy increased the abundances of *Streptococcus bovis*, *Prevotella ruminicola*, and *Ruminobacter amylophilus* at the species level. Association analysis showed that ruminal acetate was positively related to some intramuscular saturated fatty acid (SFA) contents, and *Prevotella* was significantly positively related to intramuscular total polyunsaturated fatty acid content and negatively related to intramuscular total SFA content. This study showed that high dietary energy mainly increased ruminal amylolytic and propionate-producing bacteria populations, which gave insights into how the effects of dietary energy on rumen bacteria are related to intramuscular fat fatty acids of fattening yaks.

## 1. Introduction

The yak (*Bos grunniens*) is an ancient bovine, which diverged from cattle (*Bos taurus*) 4.9 million years ago [1], distributed in the Qinghai–Tibet Plateau at altitude from 3000 to 5500 m. More than 20 million domestic yaks are raised in the world. They are important livestock, providing basic resources such as meat, milk, leather, and labor for the nomadic pastoralists of the plateau. The harsh environment of the plateau is characterized by low temperatures, forage scarcity, hypoxia, and ultraviolet radiation. The evolutionary history of yaks has equipped them with unique physiological traits that have enabled them to adapt to the extreme high-altitude environment. A previous study compared the genomes of yak and cattle and found that the efficient nutritional assimilation and energy metabolism of yaks contributed to their high-altitude adaption [1]. The rumen, harboring a diverse microbiome, is a crucial digestive organ for nutritional assimilation of ruminants. Ruminal microbes degrade carbohydrates to volatile fatty acids (VFAs) to provide 70–80% of the metabolizable energy. A recent study reported on the convergent evolution of microbes in the rumen of yaks, finding that yaks had higher VFA production, lower methane emissions, and more efficient energy-harvesting abilities compared to cattle living at low altitude [2]. Other studies characterized the profile of rumen microflora and suggested that yaks have a higher fiber-degrading capacity than low-altitude cattle [3,4,5]. However, the effects of dietary energy levels on the rumen bacterial community composition of fattening yaks are still unclear. 

The dietary energy digestion and utilization of mature ruminants are distinctly different from those monogastric animals because of ruminal bacteria fermentation. The conversion of acetate to acetyl-CoA by the cytoplasmic acetyl-CoA synthase is the dominant pathway of fat synthesis, propionate is the main precursor for gluconeogenesis in the liver, and butyrate is oxidized to provide energy for ruminants. The fat content and fatty acids in ruminant products are mainly affected by dietary nutrition and rumen bacterial metabolism [6]. Jami et al. [7] reported a strong relationship between rumen fermentation and dairy cow milk fat yield. Abbas et al. [8] found that ruminal bacteria affected intramuscular fat content of Wagyu cattle. Petri et al. [9] reported that the fatty acid profile in subcutaneous fat is related to rumen bacterial community composition. Our previous study showed that high dietary energy level significantly increased the longissimus dorsi intramuscular fat content, increased the polyunsaturated fatty acid (PUFA) content, and decreased the saturated fatty acid (SFA) content in the intramuscular fat of fattening yaks [10]. The intramuscular fat fatty acid profile is important for meat quality and human health; however, there is little information available about the correlativity between rumen bacterial populations and intramuscular fat fatty acid profiles in fattening yaks.

Therefore, the present study aimed to investigate (1) the effects of dietary energy levels on rumen bacterial populations by using 16S rDNA sequencing and (2) the relationship between rumen bacteria and intramuscular fat fatty acids in the longissimus dorsi of fattening yaks. This will give insight into how nutritional strategies regulate rumen bacteria and reveal the dominant bacteria related to intramuscular fatty acids of fattening yaks.

## 2. Materials and Methods

### 2.1. Ethics Statement

The animal experiment was performed according to the Regulation on the Administration of Laboratory Animals (2017 Revision) promulgated by Decree No. 676 of the Chinese State Council. The protocol was approved by the Institutional Animal Care and Use Committee in Sichuan Agricultural University, Sichuan, China (20200031).

### 2.2. Experimental Animals and Design

The experiment was performed on the farm (at altitude from 3200 to 3500 m) of Animal Husbandry and Veterinary Institute of Haibei Prefecture, Qinghai Province, China. Thirty healthy 3-year-old male Qinghai plateau yaks (114.57 ± 21.56 kg (mean ± SD)) were chosen and randomly assigned to three groups with 10 yaks in each group. Each group was received one of the three isonitrogenous diets with low (3.72 MJ/kg of NEmf, LE), medium (4.52 MJ/kg of NEmf, ME), and high (5.32 MJ/kg of NEmf, HE) levels of net energy for maintenance and fattening (NEmf), which is the most widely used energy indicator in the beef cattle farming of China [11]. Dietary nutrient levels of the ME group were designed to meet the Chinese Beef Cattle Raising Standard 2004 (NY/T 815–2004) recommendations for beef cattle with 150 kg body weight and 500 g/day weight gain. The concentrate-to-roughage ratio of the three diets was 3:7. Feed ingredients and nutrition levels are shown in Table 1. Briefly, yaks were fitted with ear tags, vaccinated, and treated for the external and gastrointestinal parasites. Every two yaks of the same group were penned together (3.5 × 8.0 m) and adapted to the indoor environment. After the 30-day adaptation period, a 120-day feeding trial was performed. Each pen was provided with dietary feed twice daily (09:00 and 17:00), and water was refilled once daily. Diets and fresh drinking water were offered ad libitum.

### 2.3. Sample Collection

Jugular blood samples (10 mL) of all yaks were collected on day 120 before the morning feeding. The serums were separated after centrifuge at 3500 rpm at 4 °C for 15 min and stored at −20 °C for analysis of serum biochemical and hormonal parameters. Then, six yaks in each group, which were close to the group average weight, were slaughtered humanely by captive bolt stunning and exsanguination according to the National Standard Operating Procedures of Cattle Slaughtering (GB/T 19477-2004). Digesta collected from dorsal, ventral, and caudal areas of the rumen were mixed and filtered through four-layer nylon cloth. Rumen fluid was collected in three 10-mL centrifuge tubes; one centrifuge tube of rumen fluid was immediately used to detect pH value using a pH meter (INESA, Shanghai, China), and the others were stored immediately at −80 °C for ruminal fermentation parameters and bacterial community composition analysis. The longissimus dorsi muscle samples were collected between the 12th and 13th ribs of the left side of carcass, and the intramuscular fatty acid profile was detected by using gas chromatography (Agilent Technologies, Santa Clara, CA, USA) as described in our previous study [12].

### 2.4. Serum Biochemical and Hormonal Indictors

The concentrations of serum glucose (GLU), triacylglycerols (TG), total cholesterol (TC), low-density lipoprotein cholesterol (LDL-C), and high-density lipoprotein cholesterol (HDL-C) were analyzed using an Automatic Biochemical Analyzer (SHIMADZU, Kyoto, Japan). Serum non-esterified fatty acid (NEFA) concentrations were determined by a colorimetry assay kit (Nanjing Jiancheng, Nanjing, China), referring to the manufacturer’s instructions. The concentrations of hormones, including insulin, insulin-like growth factor (IGF-1) and leptin, were detected by using the bovine-specific enzyme immunoassay kit according to the manufacturer’s instructions.

### 2.5. Ruminal Fermentation Parameters

The VFA (acetate, propionate, and butyrate) concentrations were detected by using gas chromatography (Agilent Technologies, Santa Clara, CA, USA) according to the protocol of Stewart and Duncan (1985) [13]. The ammonia concentrations were determined by using trichloroacetic acid and colorimetric method [14].

### 2.6. DNA Extraction and 16S rRNA Gene Pyrosequencing

Bacterial total genome DNA of 18 ruminal fluid samples were extracted by using the TIANamp stool DNA kit (Tiangen Biotech, Beijing, China) according to the manufacturer’s instructions. The purity and concentrations of extracted DNA were determined by agarose gel (1%) electrophoresis and the use of a micro-spectrophotometer (Thermo Scientific, Waltham, MA, USA). More than 30 μg/μL high-quality genomic DNA was extracted for each sample. The V3-V4 region of the 16S ribosomal RNA gene were amplified by using the bacterial universal primer pair (341F, 5’-CCTAYGGGRBGCASCAG-3’; 806R, 5’-GGACTACNNGGGTATCTAAT-3’) [15] with a sample-unique barcode and using the KAPA HiFi Hot Start Ready Mix PCR Kit (Kapa Biosystems, Wilmington, NC, USA). The reaction system and PCR program were set according to the manufacturer’s instructions. The PCR products were performed electrophoresis in 2% agarose gel. The GeneJET Gel Extraction Kit (Thermo Scientific, Waltham, MA, USA) was used to exact DNA fragments in the bright strips between 400–500 bp. Sequencing libraries were generated using NEB Next Ultra DNA Library Prep Kit (Ipswich, England) for Illumina, following the manufacturer’s recommendations. Then, amplicons of 18 samples were pooled in equal amounts and sequenced by using Illumina MiSeq platform (Illumina, San Diego, CA, USA) at Tinygene Biotechnology Co., Ltd. (Shanghai, China).

### 2.7. Sequencing Data Processing

The original paired-end reads were quality-filtered by Trimmomatic with the criteria described by Liu et al. [16] and merged by using FLASH. Quality reads were obtained after the ambiguous and homologous reads were screened by using Mothur V.1.33.3. The USEARCH (version 7.1) was used to cluster the operational taxonomic units (OTUs) with 97% similarity cutoff. Chimeric sequences were removed by using the UCHIME. The representative sequences of OTUs were annotated to the Silva database version 128 by using Mothur with a confidence threshold of 80%. The relative abundances of OTUs were generated, and the Chao1, abundance-based coverage estimator (ACE), and Shannon and Simpson indexes were calculated. Rarefaction curves were generated using Mothur. The Venn diagram, principal component analysis (PCA) using the Euclidean distance, and heatmap of abundance of bacterial genera were analyzed using an online tool (http://www.omicshare.com/).

### 2.8. Quantitative Real-Time PCR

The relative abundances of ruminal functional bacterial species were detected by using quantitative real-time PCR to amplify the rumen bacterial total genome DNA [17]. The primers shown in Appendix A were selected according to the published literature. The PCR was performed using the CFX96 Touch Real-Time PCR Detection System (Bio-Rad, Hercules, CA, USA) and SYBR PREMIX ExTaq kit (Takara, Dalian, China), according to the manufacturer’s instructions. Briefly, the amplification process consisted of initial denaturation at 95 °C for 10 s, followed by 40 cycles of denaturation at 95 °C for 5 s and annealing (Temperatures of primers were shown in Appendix A) for 30 s, where fluorescence was detected in each cycle. The melting curve was determined by heating the temperature from 65 to 95 °C with a 0.1 °C/s increment. PCR was performed in triplicate for all samples. The proportion of a bacterial species to the total bacterial 16S rDNA was calculated using the following equation:Relative abundance = 2^−(CT target − CT total bacteria)^, CT represented the threshold cycle.

### 2.9. Statistical Analyses 

Data were analyzed using SPSS v.19.0 (SPSS Inc., Chicago, IL, USA). The statistical differences in rumen fermentation parameters and serum indicators were analyzed by using one-way ANOVA followed by Duncan’s post hoc testing. The relative bacterial abundances were non-normally distributed, and the statistical difference was analyzed by using Kruskal–Wallis testing. Data are presented as means ± SEMs. A *p*-value of less than 0.05 was considered indicative of statistical significance. Correlation analysis between ruminal VFA concentrations or bacterial genera and intramuscular fatty acids was conducted using Spearman’s correlation analysis. A *p*-value of less than 0.05 was considered as indicative of a significant correlation.

The sequence data have been deposited in the NCBI under the BioProject Accession PRJNA509701 (https://www.ncbi.nlm.nih.gov/bioproject/?term=PRJNA509701) and SRA Accession SRP173136.

## 3. Results

### 3.1. Effects of Dietary Energy Levels on the Rumen Fermentation of Yaks

Increasing dietary energy levels significantly increased the ruminal propionate fermentation. The propionate concentration of the HE group was about 50% higher than that of the LE group (*p* < 0.05). High dietary energy level resulted in a trend of decreasing the ruminal acetate concentration (*p* = 0.057) and significantly decreased the ratio of acetate to propionate and the ammonia concentrations of yaks (*p* < 0.05) (Table 2).

### 3.2. Effects of Dietary Energy Levels on the Rumen Bacterial Community Composition of Yaks

Ruminal fluid samples from 18 yaks were sequenced and generated an average of 29,207.1 ± 205.34 (mean ± SEM) qualified reads. In total, 1270 OTUs were identified at a 97% similarity degree. The coverage indices of each sample were greater than 99%, which indicated that the sequencing data were reflected accurately (Appendix A). The ruminal microbiota of ME group had the highest Chao1, ACE, and Shannon indices (*p* < 0.05), whereas the ruminal microbiota of the HE group had the significantly lowest Chao1 and ACE indices in the three groups (*p* < 0.05) (Figure 1 and Appendix A). The PCA analysis showed that samples of HE group had distinctly different microflora from the samples of ME and LE groups. The Venn diagram showed that 1084 OTUs were shared among all groups and 9, 13, and 15 OTUs were unique in the LE, ME, and HE groups, respectively, which indicated that most of the rumen bacteria were shared by yaks fed different energy level diets (Figure 2).

In total, 16 phyla were identified from the rumen samples. The dominant bacterial phyla in the rumen of yaks (average abundance > 0.5% at least in one group) were Bacteroidetes (44.52–58.33%), Firmicutes (36.08–55.50%), Tenericutes (1.14–1.68%), Candidate division TM7 (0.66–0.79%), Lentisphaerae (0.65–0.86%), Proteobacteria (0.59–0.88%), and Synergistetes (0.35–0.55%). High dietary energy significantly increased relative abundance of Firmicutes and decreased relative abundance of Bacteroidetes (*p* < 0.05). The Firmicute population in the HE group was nearly 40% higher than that of the LE and ME groups, and the Bacteroidetes population in the HE group was about 23% lower than that of the LE and ME groups. Therefore, high dietary energy significantly increased the ratio of Firmicutes to Bacteroidetes (*p* < 0.05) (Figure 3 and Appendix A).

A total of 57 bacterial genera were identified at the genus level. The dominant genera (average abundance > 0.5% at least in one group) in the rumen of yaks included *Prevotella* (6.48–9.32%), *Succiniclasticum* (4.17–8.66%), *Saccharofermentans* (2.27–4.73), *Butyrivibrio* (1.77–2.50%), *Papillibacter* (0.77–1.31%), *Candidatus Saccharimonas* (0.62–1.17%), *Ruminococcus* (0.58–1.15%), and *Pseudobutyrivibrio* (0.25–0.55%). The ME diet resulted in a trend of increasing the abundances of *Succiniclasticum* and *Ruminococcus* and decreasing the relative abundance of *Pseudobutyrivibrio* when compared to LE group (*p* > 0.05). The ME diet also significantly increased the abundance of *Selenomonas* (*p* < 0.05). The HE diet significantly increased the abundances of *Succiniclasticum*, *Saccharofermentans*, *Ruminococcus*, *Blautia*, and *Moryella* (*p* < 0.05) and decreased the abundances of *Pseudobutyrivibrio* (*p* < 0.05) when compared to LE group (Figure 4 and Appendix A).

### 3.3. Quantitative Real-Time PCR Analysis of the Relative Abundances of Amylolytic and Fibrolytic Bacteria at the Species Level

The quantitative real-time PCR analysis revealed the relative abundance of bacterial species regulated by the dietary energy levels. Increasing dietary energy levels significantly increased the relative abundance of amylolytic bacteria, including *S. bovis*, *P. ruminicola*, and *R. amylophilus* (*p* < 0.05). However, high dietary energy resulted in a trend of decreasing the relative abundance of fibrolytic bacterial species such as *R. albus* and *B. fibrisolven* (*p* > 0.05) (Table 3).

### 3.4. Effects of Dietary Energy Levels on the Serum Biochemical and Hormonal Indicators of Yaks

As the dietary energy levels increased, the serum glucose concentration significantly increased and the serum NEFA concentration significantly decreased (*p* < 0.05). Serum triglyceride and LDL-C concentrations of yaks in the HE and ME groups were significantly higher than those of LE group (*p* < 0.05). Serum total cholesterol and HDL-C concentrations of yaks in the HE group were significantly higher than those of the LE group (*p* < 0.05). Increasing dietary energy levels significantly increased the serum IGF-1 concentrations of yaks (*p* < 0.05). The serum leptin concentration of the HE group was significantly higher than that of the ME and LE groups (*p* < 0.05), and the serum insulin concentration of HE group was significantly higher than that of the LE group (*p* < 0.05) (Figure 5).

### 3.5. Association Analysis between Ruminal Bacterial Genera and Intramuscular Fat Fatty Acids of Yaks

The association analysis showed that ruminal VFA concentrations and predominant bacterial genera abundances were significantly related to the intramuscular fatty acid profile (Figure 6). The acetate concentration was positively associated with intramuscular C13:0, C22:0, and C18:3 contents (*p* < 0.05), and propionate concentration was positively associated with intramuscular C15:0 (*p* < 0.05). Interestingly, the relative abundance of *Prevotella* was negatively related to the intramuscular C14:0 (*p* < 0.05), C15:0 (*p* < 0.01), and total SFA (*p* < 0.05) contents, whereas the *Prevotella* abundance had a significantly positive relationship with the intramuscular C18:1 and total MUFA contents (*p* < 0.05). The abundances of *Candidatus Saccharimonas*, *Christensenella*, and *Moryella* were positively related to intramuscular C15:0 and C17:0 contents (*p* < 0.05). The abundance of *Oribacterium* was positively related to intramuscular C22:0 content (*p* < 0.01) and negatively related to intramuscular C16:0 content (*p* < 0.05). The *R. albus* abundance correlated positively with intramuscular C18:0 contents (*p* < 0.05) and correlated negatively with intramuscular C24:0 contents (*p* < 0.05).

## 4. Discussion

Previous studies have reported that dietary calorie contents and sources affect the rumen microbiota and fatty acid profile in milk and meat of dairy cows and beef cattle [18]. The yak rumen, colonized by a mass of microorganisms, plays crucial roles in the dietary energy harvest and adaption to the plateau environment. To our knowledge, this was the first study to investigate how dietary energy levels regulate the rumen microbiota and reveal the specific bacteria related to the intramuscular fatty acid profile of fattening yaks by using the 16S rDNA sequencing technique. 

### 4.1. Effects of Dietary Energy Levels on the Rumen Fermentation and Bacterial Community Composition of Yaks

VFAs are the important sources of metabolizable energy for ruminants, of which acetate, propionate, and butyrate account for about 95% of total VFAs. Acetate is the dominant precursor for synthesizing fat in the mammary gland and adipose tissue of ruminants. Propionate is converted into glucose through the gluconeogenesis pathway in the liver. Butyrate is mainly used to provide energy through the oxidative metabolism pathway. A high-energy diet, which contained plentiful rapidly fermentable carbohydrates such as grain starch, significantly increased the propionate concentrations in the rumen of yaks. The results suggested that a high-energy diet may significantly promote the growth of propionate-producing bacteria and increase the energy harvest from the rumen ecosystem. The dietary nitrogen was degraded to ammonia and used for microbial protein synthesis by bacterial fermentation. Ruminal ammonia concentration represents the utilization efficiency of dietary nitrogen by the bacteria. The results showed that high dietary energy significantly decreased ruminal ammonia concentration, which suggested that increasing the dietary energy level also facilitated the utilization efficiency of dietary nitrogen in the rumen. 

The 16S rRNA gene sequencing of rumen fluids was performed to investigate the effects of dietary energy levels on the rumen bacterial community composition of yaks. The rarefaction curve (Appendix A) shows that the sequencing depth of this study was sufficient to reveal the ruminal bacterial diversity. The results found that a high-energy diet reduced the rumen bacterial richness and diversity, which suggested that high contents of dietary grain starch and lipid were disadvantageous to some ruminal bacteria. Filippo et al. [19] also found that a high-fiber diet enriched the microbiome and a nutrient-rich diet decreased the microbial richness and biodiversity. The PCA analysis found that the rumen microflora of the HE group was significantly different from that of the ME and LE groups, which suggested the high dietary energy more effectively changed the ruminal bacterial composition of yaks.

At the phylum level, it was found that *Bacteroidetes* (44.52–58.33%) and *Firmicutes* (36.08–55.89%) are the most dominant phyla in the rumen of yaks; these results are similar to those found for the dominant ruminal phyla of grazing yaks and other ruminants [3,20,21,22]. The results showed that feeding a high-energy diet significantly increased the abundance of *Firmicutes* and the ratio of *Firmicutes* to *Bacteroidetes* (F/B). Previous studies suggested that high *Firmicutes* and low *Bacteroidetes* populations indicated a high energy-harvesting ability of the host, and the reduction of F/B was strongly associated with body weight loss [18]. A strongly positive relationship was also found between the ruminal F/B and milk-fat yield of dairy cow [7]. Liu et al. [23] found that high-grain feed significantly increased the proportion of *Firmicutes* in the rumen of goats. Therefore, high-energy diet may stimulate body fat deposition of yaks through changing the rumen bacterial composition and increasing energy extraction from the rumen ecosystem.

At the genus level, it found that *Prevotella* was the most dominant bacterial genus in the rumen of yaks. Previous studies have proven that *Prevotella* was the most abundant bacterial genus in the rumen of dairy cow and beef steers and calves, showing its proteolytic action [24]. The results showed that *Succiniclasticum* was the second most dominant bacterial genus in the rumen of yaks, and it significantly increased in abundance following the increase in dietary energy levels. Ruminal *Succiniclasticum*, which was firstly isolated from dairy cow in 1995, plays crucial roles in energy harvest through converting succinate into propionate [25]. The *Saccharofermentans* are typical amylolytic bacteria [26], and the HE diet with high starch content significantly increased the relative abundance of *Saccharofermentans* in the rumen. The abundance of *Ruminococcus* was increased following the increase in dietary energy levels. The *Ruminococcus* population was higher in the yaks supplemented with concentrate compared to grazing yaks [27]. The results suggested that improving dietary nutrition increased the *Ruminococcus* population in the rumen of yaks. The relative abundance of *Blautia* increased about three times in the HE group when compared with the ME and LE groups. The *Blautia* mainly ferment polysaccharides and utilize H_2_ and CO_2_ to produce acetate, which competes with hydrogen in the methane synthesis pathway. Acetate, methane, and hydrogen sulfide synthesis are the main ways of hydrogen removal in the gut microbial community, and they also affect the efficiency of energy harvest from dietary nutrients [28]. Our results suggest that high dietary energy may increase the energy extraction in the rumen ecosystem partly through increasing the *Blautia* population to decrease methane production. The highest relative abundance of *Selenomonas* in the ME group and lowest relative abundance in the HE group suggest that a moderate increase in dietary energy stimulates *Selenomonas* growth, while high dietary energy may more effectively increase the abundance of other starch-utilizing bacteria. *Moryella* was another bacterial genus regulated by dietary energy densities in this study. A previous study also found a higher abundance of *Moryella* in the rumen of cattle fed a high-energy diet than in that of cattle fed a low-energy diet [29], but the specific functions of *Moryella* need to be studied further. On the other hand, *Pseudobutyrivibrio* was identified as a group of ruminal fibrolytic bacteria [30]. The high-energy diet had lower fiber content, so yaks in the HE group had the significantly lowest relative abundance of *Pseudobutyrivibrio*. Therefore, the results suggested that high dietary energy potentially stimulated the growth of ruminal amylolytic bacteria growth and was disadvantageous to the growth of fibrolytic bacteria.

For the in-depth detection of the bacterial species regulated by dietary energy levels, quantitative real-time RCR was used to verify the variation in relative abundance of amylolytic and fibrolytic bacteria at the species level. *P. ruminicola*, *S. bovis* and *R. amylophilus* are main amylolytic and propionate-producing bacteria in the rumen. A high-energy-level diet significantly increased the relative abundance of these bacteria and stimulated propionate fermentation in the rumen. Generally, cattle have significantly higher abundances of amylolytic bacteria when their diets are changed from high-forage to high-grain [31]. The *F. succinogenes*, *B. fibrisolven*, *R. albus*, and *B. fibrisolven*, which secrete cellulase and hemicellulose, are considered as the dominant fibrolytic bacterial species in the rumen. As the fiber contents decrease in the high-energy diet, the relative abundances of fibrolytic bacterial species such as *R. albus* and *B. fibrisolven* exhibited a decreasing trend. A previous study also reported that ruminal fibrolytic bacterial populations are positively related to dietary fiber contents and negatively related to dietary starch contents [32]. Therefore, these results suggest that a high-energy diet increased the relative abundance of amylolytic bacteria populations and propionate production to enhance the energy harvest in the rumen ecosystem. Dietary energy is an aggregative indicator to reflect the contents of starch, lipid, protein, and fiber. In this study, the dietary corn starch content increased and fiber contents decreased as the dietary energy increased. The rumen bacterial composition and fermentation were mainly affected by the proportions of structured and unstructured carbohydrates in the different energy diets.

### 4.2. Effects of Dietary Energy Levels on the Serum Energy Metabolite and Hormone Concentrations of Yaks

Glucose is the main energetic substance of animals, and blood glucose concentration represents the energy metabolism status of the body. Blood glucose of ruminants is dominantly produced through gluconeogenesis from propionate in the liver [33]. The high-energy diet, which contained more corn starch, produced more propionate by the ruminal microorganism fermentation and increased the blood glucose concentration of yaks. Serum TG, TC, LDL-C, HDL-C, and NEFA are the typical energy and lipid metabolites. TG is the most important substance for storing energy in the body, and NEFA is the catabolite product of TG. Concentrations of serum TG and NEFA represent the fat metabolism and energy homeostasis [34]. The significantly high serum TG, TC, LDL-C, and HDL-C concentrations and low NEFA concentration suggested increased energy harvest from the rumen ecosystem and increased fat accumulation in the yaks fed the high-energy diet. 

Energy metabolism is also regulated by several hormones, such as IGF-1, insulin, and leptin. These serum hormone concentrations are affected by the dietary energy levels. Leptin is secreted by adipose tissue, and serum leptin concentration is positively associated with body fat content. Starvation and malnutrition decrease the serum leptin concentration of animals [35]. A high serum leptin concentration suggests that high dietary energy is stimulating the energy metabolic rate [36] and allowing more fat to accumulate in the yaks. Insulin regulates the synthesis and utilization of blood glucose, glycogen, and fat. IGF-1 stimulates muscle to absorb and utilize the blood glucose. Serum insulin and IGF-1 concentrations rise following the increased nutrition intake and are positively related to serum glucose concentrations [37]. Our results suggest that high dietary energy stimulates insulin and IGF-1 secretion and facilitates the energy metabolism of yaks. 

### 4.3. Association Analysis between Ruminal Bacterial Genera and Intramuscular Fat Fatty Acids of Yaks 

Acetate is the major precursor of fat synthesis, and propionate is crucial for the gluconeogenesis of ruminants. This study found that ruminal acetate concentration was positively related to intramuscular C13:0, C22:0, and C18:3 contents, and propionate concentration was positively related to intramuscular C15:0 content. Dietary lipids mainly provide fatty acids for the fat synthesis of ruminants and affect the rumen bacterial composition. Table 1 shows that high-energy diet had high lipid contents, which potentially affected the correlation between the bacterial population and the intramuscular fatty acid composition. Ruminal bacteria are crucial for fatty acid composition in the products of ruminants because of biohydrogenation and the provision of precursors for lipogenesis. Gillis et al. [38] reported that 68–84% of unsaturated fatty acids (UFAs) in the diet were converted to SFA by bacterial biohydrogenation. Notably, *Prevotella* abundance was strongly positively related to intramuscular PUFA content and negatively related to intramuscular SFA content. This result is due to *Prevotella* mainly utilizing starch and playing a crucial role in energy harvesting in the rumen ecosystem [39], in addition to potentially providing precursors for UFA synthesis. Abundances of *Candidatus Saccharimonas*, *Christensenella*, and *Moryella* were positively related to intramuscular C15:0 and C17:0 contents, and these bacterial genera are potentially crucial for biohydrogenation in the rumen. Other studies reported *Saccharimonas*, *Christensenella*, and *Moryella* mainly affect the ruminal energy metabolism and are related to adipose metabolism.

## 5. Conclusions

This study investigated dietary energy levels regulating the rumen microflora of yaks by using 16S rRNA gene sequencing. It showed that high dietary energy significantly increased the abundance of *Firmicutes* and ratio of *Firmicutes* to *Bacteroidetes* at the phylum level. The high dietary energy also increased the abundances of amylolytic bacteria, such as *Succiniclasticum* and *Saccharofermentans*, and decreased the cellulolytic *Pseudobutyrivibrio* population at the genus level. Quantitative real-time RCR verified the increase in ruminal amylolytic bacteria and decrease in fibrolytic bacteria at the species level as the dietary energy level rose. High dietary energy significantly stimulated ruminal propionate fermentation and energy harvest. The association analysis suggested that rumen VFA and bacteria were strongly related to intramuscular fat fatty acids in the longissimus dorsi of fattening yaks. Interestingly, this study found that *Prevotella* abundance was positively related to PUFA content and negatively related to SFA in the *longissimus dorsi*. These rumen microbes and intramuscular fatty acid variations were potentially affected by the different contents of energy ingredients among the three diets, such as lipids, starch, and fiber. Future studies are needed to research the mechanisms of the rumen ecosystem regulating deposition of fatty acids in the muscle of yaks. Regardless, this study showed how dietary energy levels regulate the rumen microflora and revealed the specific bacteria related to intramuscular fatty acids of fattening yaks.

## Figures and Tables

**Figure 1 animals-10-01474-f001:**
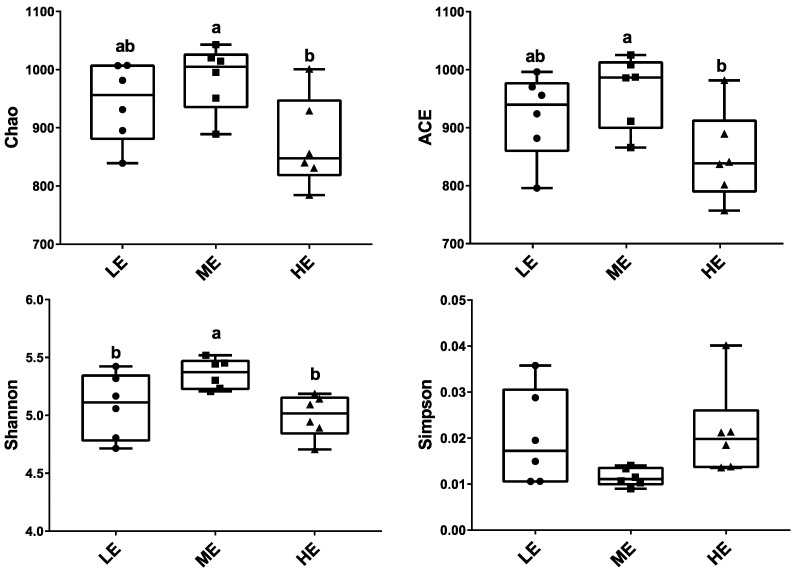
Effects of dietary energy levels on the rumen bacterial α diversity of yaks. Each group had 6 yaks. Different lowercase letter superscripts represent significantly different (*p* < 0.05).

**Figure 2 animals-10-01474-f002:**
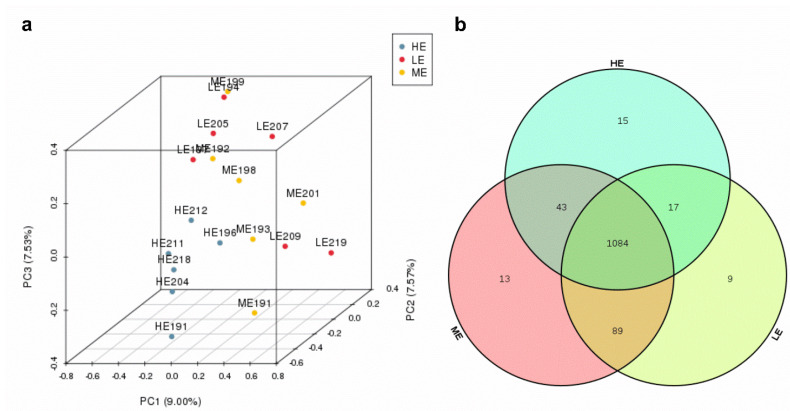
Effects of dietary energy levels on the rumen bacteria of yaks: (**a**) PCA analysis and (**b**) Venn diagram analysis. Each group had 6 yaks.

**Figure 3 animals-10-01474-f003:**
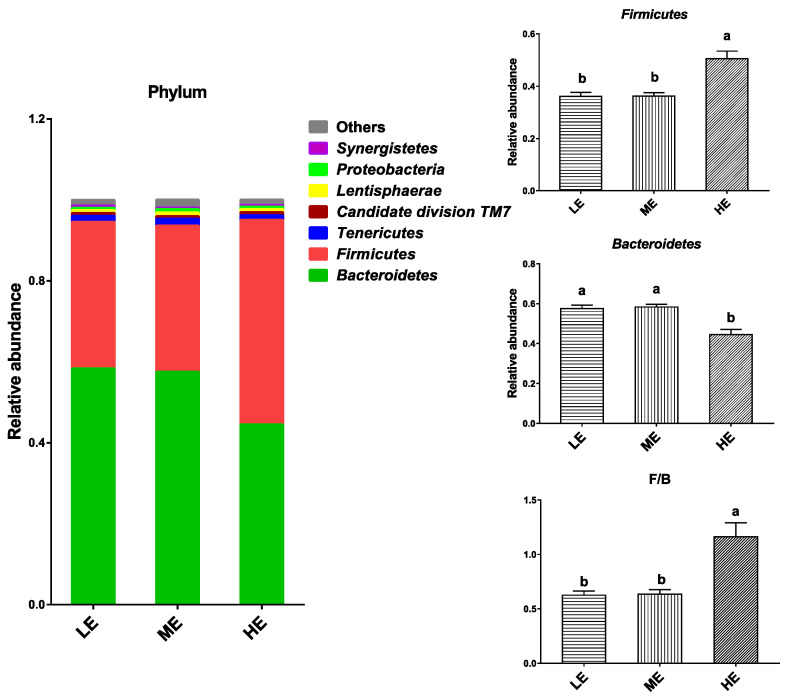
Effects of dietary energy levels on the yak rumen bacterial composition at the phylum level. Each bar and color represent the average relative abundance of each phylum, and the 7 most abundant taxa are shown. Each group had 6 yaks. Values are means ± SEMs. Bars with different lowercase letter superscripts are significantly different (*p* < 0.05).

**Figure 4 animals-10-01474-f004:**
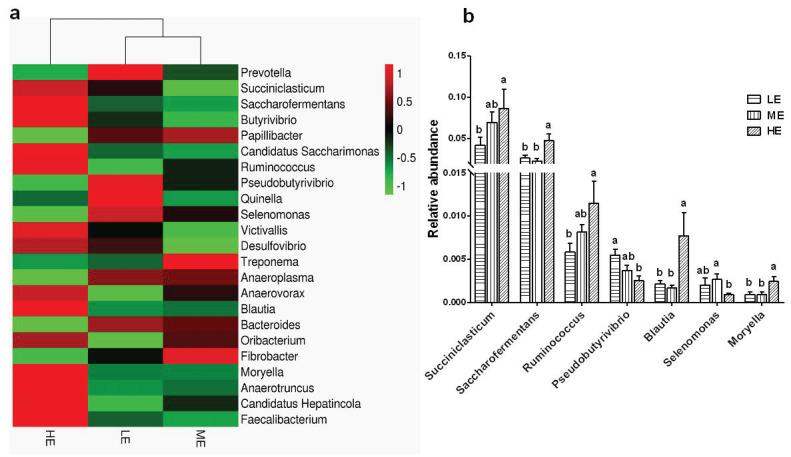
Effects of dietary energy levels on the yak rumen bacterial composition at the genus level. (**a**) Heatmap showing the average relative abundance of each genus taxon (relative abundance > 0.1%) in different groups. The Y-axis represents different bacterial genera, and the X-axis represents different groups. The relative abundances of bacterial genera are represented by color intensity according to the scale provided. (**b**) Bar graph showing the significantly fluctuant bacteria at the genera level. Values are means ± SEMs. Bars with different lowercase letter superscripts are significantly different (*p* < 0.05). Each group had 6 yaks.

**Figure 5 animals-10-01474-f005:**
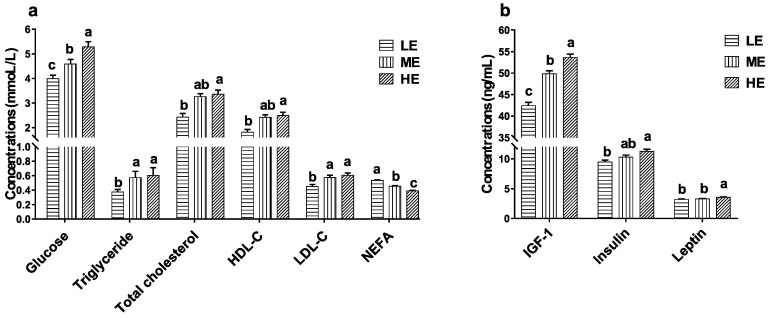
Effects of dietary energy levels on the serum biochemical indicator concentrations (**a**) and hormone concentrations (**b**) of yaks. Each group had 10 yaks. Values are means ± SEMs. Bars with different lowercase letter superscripts are significantly different (*p* < 0.05).

**Figure 6 animals-10-01474-f006:**
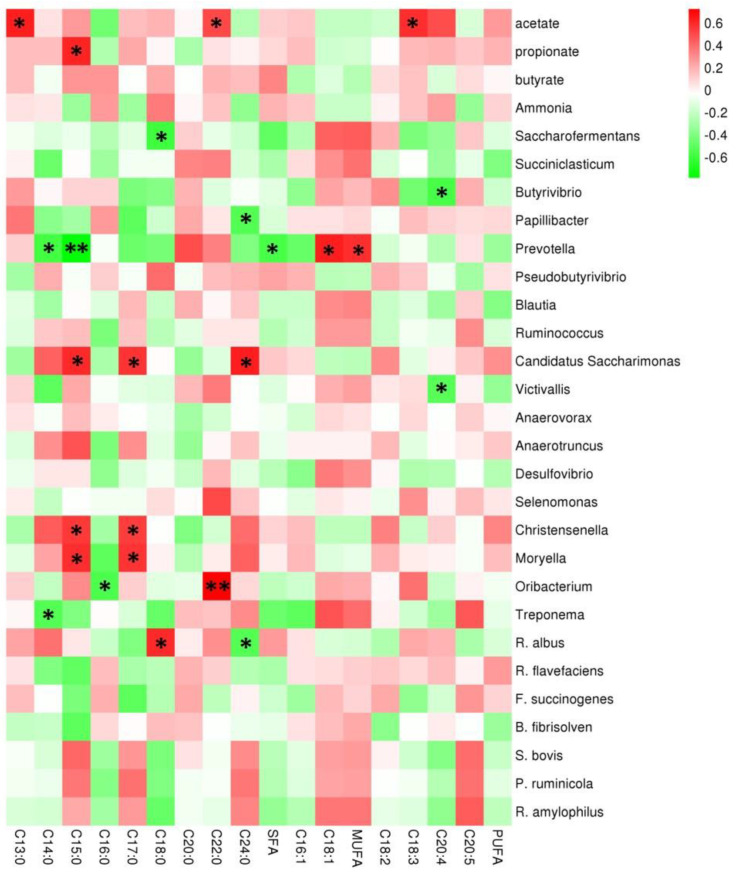
The correlations between the rumen volatile fatty acids (VFAs) or predominant bacterial genera and intramuscular fat fatty acids in the longissimus dorsi of fattening yaks. The Y-axis represents different VFAs and bacterial genera, and the X-axis represents the different intramuscular fat fatty acids. Color intensity represents *p*-values of correlation; ∗ *p* < 0.05 and ∗∗ *p* < 0.01.

**Table 1 animals-10-01474-t001:** Ingredients and nutrient levels of experimental diets (air-dry basis, %).

Items	Treatments ^a^		
	LE	ME	HE
Ingredients			
Corn	2.20	15.84	22.75
Wheat bran	22.03	6.62	0.40
Soybean meal	2.95	3.38	3.37
Rapeseed meal	1.25	2.25	1.10
Calcium hydrophosphate	0.00	0.59	1.20
Calcium carbonate	0.64	0.39	0.25
Sodium chloride	0.30	0.30	0.30
Sodium bicarbonate	0.30	0.30	0.30
Choline chloride	0.03	0.03	0.03
Premix (trace minerals and vitamins) ^b^	0.30	0.30	0.30
Oaten hay	60.00	47.50	30.00
Highland barley distiller grains	10.00	22.50	40.00
In total	100.00	100.00	100.00
Nutrition levels			
NEm (MJ/kg) ^c^	5.72	6.21	6.63
NEg (MJ/kg) ^c^	2.87	3.46	3.98
NEmf (MJ/kg) ^c^	3.72	4.52	5.32
Crude protein, CP (%)	12.57	12.57	12.57
Ether extract, EE (%)	4.81	5.56	6.43
Crude fiber, CF (%)	21.35	17.92	13.62
Neutral detergent fiber, NDF (%)	47.64	43.43	41.19
Acid detergent fiber, ADF (%)	26.27	25.34	24.58
Calcium (%)	0.60	0.59	0.60
Phosphorus (%)	0.40	0.39	0.40

^a^ LE = low-energy-level diet, ME = medium-energy-level diet, HE = high-energy-level diet. ^b^ The premix provided trace minerals and vitamins to the diet, containing 4400 IU VA, 550 IU VD, 110 IU VE, 0.10 mg Co, 10 mg Cu, 0.50 mg I, 30 mg Mn, 0.20 mg Se, 30 mg Zn, and 50 mg Fe per kilogram. ^c^ NEm is net energy for maintenance. NEg is net energy for gain. NEmf is net energy for maintenance and fattening. The NEmf were calculated according to NEg and NEm following [11]. The other nutrient compositions were detected according to the methods of Association of Official Analytical Chemists (AOAC).

**Table 2 animals-10-01474-t002:** Effects of dietary energy levels on the rumen fermentation parameters of yaks.

Items	Groups			SEM	*p*-Value
	LE	ME	HE		
pH	6.75	6.70	6.67	0.031	0.641
Acetate (mmol/L)	47.23	45.94	44.74	0.639	0.057
Propionate (mmol/L)	9.48 ^c^	12.00 ^b^	14.19 ^a^	0.314	0.019
Butyrate (mmol/L)	2.74	2.58	2.65	0.230	0.66
Acetate/Propionate	5.01 ^a^	3.83 ^b^	3.18 ^c^	0.220	0.002
Ammonia (mg/dL)	11.32 ^a^	10.46 ^b^	9.33 ^c^	0.205	0.000

SEM: standard error of the mean. Each group had 6 yaks. Data with different lowercase letter superscripts within the same row are significantly different (*p* < 0.05).

**Table 3 animals-10-01474-t003:** Effects of dietary energy levels on the relative abundance of amylolytic and fibrolytic bacteria at the species level in the rumen of yaks.

Items	Groups			SEM	*p*-Value
	LE	ME	HE		
*R. albus* (×10^−2^)	0.28	0.24	0.22	0.010	0.067
*R. flavefaciens* (×10^−3^)	0.52	0.51	0.48	0.011	0.230
*F. succinogenes* (×10^−2^)	0.92	0.88	0.87	0.014	0.329
*B. fibrisolven* (×10^−2^)	0.68	0.64	0.61	0.014	0.102
*S. bovis* (×10^−4^)	0.29 ^c^	0.49 ^b^	0.90 ^a^	0.063	0.000
*P. ruminicola* (×10^−2^)	0.17 ^c^	0.32 ^b^	0.53 ^a^	0.037	0.000
*R. amylophilus* (×10^−3^)	0.71 ^c^	0.89 ^b^	1.09 ^a^	0.039	0.000

SEM: standard error of the mean. Each group had 6 yaks. Data with different lowercase letter superscripts within the same row are significantly different (*p* < 0.05).

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
