# Peer review of "Dietary Energy Levels Affect Rumen Bacterial Populations that Influence the Intramuscular Fat Fatty Acids of Fattening Yaks (Bos grunniens)"

_animals, 2020, doi:10.3390/ani10091474_

Round 1
Reviewer 1 Report
The fact that yaks are used in the current manuscript already represents a novelty. I believe the work warrants consideration for publication; however, there are a few modifications that could be done to potentially improve the quality of the final manuscript.
My first recommendation to authors is to either seek help of a native English speaker, or seek professional help to proof read the final material. As it is, the English language utilized is clear and the messages are there, but the quality can be considerably improved with minor touches along the manuscript.
Examples: Line 17 (Simple Summary): “Yak, belonged to bovine, is the….” Why the use of past tense? Probably just a typo, but there are a few along the manuscript.
Line 52 (Introduction): “diverged with cattle..” Diverged with or from? Maybe another typo?
Line 104: Replace the word “were” for “was” - “ The concentrate-to-roughage ratio…. “was".
In general the work is easy to follow, the methodologies are sound and clear. Despite this, I list some suggestions and raise questions in an attempt to help the authors improve the overall quality of their manuscript:
Lines 35 to 36 (Abstract): “with low, medium and high energy levels”. I suggest you indicate what are those levels in either total digestible nutrients, or fermentable energy. Up to you, but saying low to high does not say much.
Line 61 to 62: “Ruminal microbes degrade carbohydrates to volatile fatty acids (VFA) to provide 70~80% metabolizable energy”. Please add reference or be specific. I am assuming you are referring to work done with cattle or are you referring to ruminants in general?
Line 110: Table 1. Note: Your diets have very distinct amount of lipids, and that will have a tremendous impact on rumen microbes and fatty acid composition of the intramuscular fat. What I am trying to say, is to be careful with conclusions because what is varying here is not only the level of energy, but the source of energy ingredients (fat vs starch for example).
Line 296 (Discussion): Why do you start with a reference from studies on human and mice? There are a range of studies done with ruminants. How these species extract their energy from fibrous materials is intrinsic of their capacity to live with a symbiotic relationship with microbes at the rumen level. As you say in Lines 299 to 301, “this was the first study to investigate … fattening yaks”. I believe you are probably right when you refer to yaks, but there are plenty of work showing the interference of energy levels AND source on rumen microbiome and fatty acid profile in ruminants. I highly recommend you read at least one of the papers on the effects of both carbohidrates by decreasing the rumen pH or PUFA on fat synthesis.
For example: Bauman, D. E., K. J. Harvatine, and A. L. Lock. 2011. Nutrigenomics, rumen-derived bioactive fatty acids, and the regulation of milk fat synthesis. Annual Review of Nutrition 31(1):299-319. doi: 10.1146/annurev.nutr.012809.104648.
Or alternatively one more specific to intramuscular fat: Smith, S. B., H. Kawachi, C. B. Choi, C. W. Choi, G. Wu, and J. E. Sawyer. 2008. Cellular regulation of bovine intramuscular adipose tissue development and composition. Journal of Animal Science:1-38. doi: 10.2527/jas.2008-1340
Note: Your reference list is not on alphabetical order, what made it hard for me to check if you had some of the main references on a particular topic.
Line 414 to 428 (Conclusions): Again, the conclusions are OK but it is important to highlight the possible confounding effects of level and source. Perhaps leave as is, but I suggest that you comment about it at least during the discussion of your material.
Reviewer 2 Report
The document is an interesting study about the effect of energy dietary level on rumen microbial population and the relationship of these microorganisms and intramuscular fat fatty acids of Yaks. The authors present the subject with a good introduction. Material and methods are well described, although they should include a few lines to explain the analyse of the chemical composition of the diets (see the comments below). Results are fine described and structured as the discussion section where results are well argued. Finally, the conclusions are supported by the results.
Simple summary: Abbreviations are not permitted in this section.
Line 22: …dietary “energy” levels…?
Abstract: Authors should eliminate the abbreviations that are not used later in this section.
Keywords: I suggest to the authors to include the word “16S rDNA sequencing”
Line 62: “A” recent study…
Line 64: “production” instead of “products”
Line 74: The correct format of the cite is Jami et al. [7]. Please, check the rest of the cites.
Material and methods: authors should check the numbering of the subsections.
Authors should explain, briefly, how did they analyse the chemical composition of the diet.
Line 138: Authors should define IGF-1.
Line 168: Authors should define ACE.
Line 170: Authors should describe how was the PCA constructed? Based in which distance.
Statistical analysis: As authors say in this section, “The bacterial abundances were non-normal distribution”, so the Pearson’s correlation analysis is not suitable for this kind of data. A Spearman’s correlation analysis should be realized.
Table 2: Authors should check the format. There are extra words in bold.
Line 213: Authors should replace “microflora” by “microbiota”.
Lines 227-231: Authors should replace “Firmicute” and “Bacteroidete” by “Firmicutes” and “Bacteroidetes”
Table 3: Authors should check the format. There are extra words in bold.
The results in the subsection 3.5 should be modified if the correlation analysis test is changed to the Spearman’s one.
Line 304: Authors should replace “microflora” by “microbiota”.
Subsection 4.3 should be revised if the correlation analysis test is changed to the Spearman’s one.
Lines 427-430: These two phrases should be modified if the Spearman correlation test is done.
Reviewer 3 Report
Dear authors,
In this study, authors studied “Dietary energy levels affected rumen bacterial populations and the correlation with intramuscular fat fatty acids of fattening yaks (Bos grunniens)”. The manuscript is well written, but I have some comments to this manuscript.
- For quantification of some specific bacterial species, you used relative quantification methods. But my concern is that the abundance of total bacteria was also may be influenced by the dietary energy level. Did you see differences in the abundance of total bacteria? In the results and discussion part, it has to be clearly mentioned that you only studied the relative abundance not the absolute abundance.
- Do you have the total VFA concentrations? Do you see any differences.
Some specific comments
- Line 22: dietary energy levels
- Line 24: ruminal propionate concentration
- Line 34: Mention number of yaks used- in this study, ** yaks were randomly ….
- Line 40: Blautia populations relative abundance: In 16S rRNA sequencing only the relative abundance of bacteria was studied, not the growth.
- Table 1 Crude Fiber
- Line 180: total bacterial 16S rRNA
- Line 185: the bacterial relative abundance
- Line 208: the ruminal microflora of ME group had the highest Chao 1, ACE…… This sentence was not true. I do not see differences between LE and ME.
- Line 235-244: please mention all genus name in italics.
- Line 226-Line 227: Bacteroidetes, Firmicutes
- Figure 4: Genus names has to be in italics
- Line 255-259: you only studied the relative abundance of specific bacteria, not the abundance. Please change the abundance to relative abundance. Also change the genus names in italics.
- Line 282: Prevotella relative abundance
- Line 308: promoted the VFAs fermentations: How did you know this? You do not have the total VFA concentrations in each group of animals.
- Line 319: The results…………. How did you know the total bacterial abundance was reduced in HE diet? Do you have the total bacterial abundance in each dietary groups?
Round 2
Reviewer 1 Report
The manuscript has improved significantly. I thank the authors for the effort to address the concerns that I had originally risen.
As now mentioned in your manuscript, not only the energy content but also energy source have tremendous impact on rumen microbiota and fatty acid profile in cattle. It is not expected to be much different in yaks.
Looking forward to seeing your final version published.
Kind regards,
Reviewer 1
Author Response
Dear Reviewer,
Special thanks to you for your careful and valuable comments again. We have uploaded a new version of the revised manuscript. We hope the correction below can meet with approval.
- As now mentioned in your manuscript, not only the energy content but also energy source have tremendous impact on rumen microbiota and fatty acid profile in cattle. It is not expected to be much different in yaks.
We have added some sentences about the energy source and ingredient affecting rumen bacterial composition and fatty acid profile in the discussion and conclusion on line 323, line 380-384, line 410-413 and 436-437.
Please check if this revision is OK, thank you so much!
Reviewer 2 Report
Thank you for your answers. Everything in the document is correct for publishing, but there is a question that was not clearly answered. In relation with the PCA, which distance was used to build it? Bray-Curtis, unifrac, ....
Author Response
Dear Reviewer,
we are so sorry for leaving out this question. For building PCA, we used the Euclidean distance, and we added this in the Materials and Methods on line 171. Thanks again for your kind mention. We hope this correction can meet with approval.